# Neurotrophic Factor BDNF, Physiological Functions and Therapeutic Potential in Depression, Neurodegeneration and Brain Cancer

**DOI:** 10.3390/ijms21207777

**Published:** 2020-10-21

**Authors:** Luca Colucci-D’Amato, Luisa Speranza, Floriana Volpicelli

**Affiliations:** 1Department of Environmental, Biological and Pharmaceutical Sciences and Technologies, University of Campania “Luigi Vanvitelli”, 81100 Caserta, Italy; 2InterUniversity Center for Research in Neurosciences (CIRN), University of Campania "Luigi Vanvitelli", 80131 Naples, Italy; 3Department of Neuroscience, Albert Einstein College of Medicine, New York, NY 10461, USA; luisa.speranza@einsteinmed.org; 4Department of Pharmacy, School of Medicine and Surgery, University of Naples Federico II, 80131 Naples, Italy; floriana.volpicelli@unina.it

**Keywords:** BDNF, miRNAs, neurogenesis, synaptic plasticity, depression, neurodegeneration, glioblastoma

## Abstract

Brain-derived neurotrophic factor (BDNF) is one of the most distributed and extensively studied neurotrophins in the mammalian brain. BDNF signals through the tropomycin receptor kinase B (TrkB) and the low affinity p75 neurotrophin receptor (p75NTR). BDNF plays an important role in proper growth, development, and plasticity of glutamatergic and GABAergic synapses and through modulation of neuronal differentiation, it influences serotonergic and dopaminergic neurotransmission. BDNF acts as paracrine and autocrine factor, on both pre-synaptic and post-synaptic target sites. It is crucial in the transformation of synaptic activity into long-term synaptic memories. BDNF is considered an instructive mediator of functional and structural plasticity in the central nervous system (CNS), influencing dendritic spines and, at least in the hippocampus, the adult neurogenesis. Changes in the rate of adult neurogenesis and in spine density can influence several forms of learning and memory and can contribute to depression-like behaviors. The possible roles of BDNF in neuronal plasticity highlighted in this review focus on the effect of antidepressant therapies on BDNF-mediated plasticity. Moreover, we will review data that illustrate the role of BDNF as a potent protective factor that is able to confer protection against neurodegeneration, in particular in Alzheimer’s disease. Finally, we will give evidence of how the involvement of BDNF in the pathogenesis of brain glioblastoma has emerged, thus opening new avenues for the treatment of this deadly cancer.

## 1. Introduction

The neurotrophin BDNF is one of the most studied and well characterized neurotrophic factors in the CNS. It regulates many different cellular processes involved in the development and maintenance of normal brain function by binding and activating the TrkB, a member of the larger family of Trk receptors. In the brain, BDNF is expressed by glutamatergic neurons [1], glial cells, such as astrocytes isolated from the cortex and hippocampus, but not from the striatum [2], and microglia [3]. During embryogenesis, BDNF–TrkB signaling promotes the differentiation of cortical progenitor cells and later promotes differentiation of cortical progenitor cells into neurons (i.e., neurogenesis) [4]. Several lines of evidence also suggest that the BDNF/TrkB signaling is involved in adult neurogenesis in the hippocampus with differing effects in the dentate gyrus (DG) and subventricular zone (SVZ) [5]. Adult neurogenesis in the dentate gyrus is enhanced by voluntary exercise, exposure to an enriched environment, and chronic antidepressant administration. Recently, it has also been proposed that caloric restriction and intermittent fasting in particular, appears to positively modulate hippocampal neurogenesis and BDNF [6]. The connection between BDNF and the modulation of hippocampal neurogenesis by external stimuli is a topic that has been extensively studied in recent years [7]. It has been demonstrated that voluntary physical exercise, like an enriched environment, increases expression of BDNF in the hippocampus [8], as well as hippocampal neurogenesis [9]. Physical exercise is one particularly effective strategy for increasing circulating levels of BDNF [10,11] and improving brain function [12,13,14].

In addition, studies also show that BDNF is an important regulator of synaptic transmission and long-term potentiation (LTP) in the hippocampus and in other brain regions. The effects of BDNF on LTP are mediated by the TrkB receptor. Especially in the hippocampus, this neurotrophin is thought to act on both the pre- and post-synaptic compartments, modulating synaptic efficacy, either by changing the pre-synaptic transmitter release, or by increasing post-synaptic transmitter sensitivity [15,16] to induce a long-lasting increase in synaptic plasticity. Additionally, converging data now suggest a role for BDNF in the pathophysiology of brain-associated illnesses. Deficits in BDNF signaling are reported to contribute to the pathogenesis of several major diseases, such as Huntington’s disease, Alzheimer’s disease (AD), depression, schizophrenia, bipolar, and anxiety disorders. Thus, manipulating the BDNF signaling may present a viable approach to treat a variety of neurological and psychiatric disorders. BDNF protein is also detectable outside of the nervous system in several non-neuronal tissues, such as in endothelial cells [17,18], cardiomyocytes [19], vascular smooth muscle cells [17], leukocytes [20], platelets [21,22], and megakaryocytes [19]. Therefore, it may also be involved in cancer, angiogenesis, reduction of glucose production from the liver [23], and in the uptake of glucose in peripheral tissues (see [24] for review). In addition, BDNF promotes the development of neuromuscular synapses and is required for fiber-type specification, suggesting a potential role as a therapeutic target in muscle diseases [25]. In this review, first we examine the currently known mechanisms of BDNF signaling, information essential for the creation of BDNF-based therapeutics. Next, we focus on the effects of antidepressants on BDNF-mediated plasticity. Additionally, we highlight the function of BDNF as a potent factor capable of conferring protection against neurodegeneration. Finally, we touch on the newly emerging role of BDNF in the pathogenesis of brain gliomas.

## 2. The Human BDNF Gene: Transcripts and Variants

### 2.1. BDNF Transcripts

The *BDNF* gene codes for a neurotrophin that is highly expressed in the CNS [26]. At the beginning of 2000s, the only data available about the structure and regulation of the *BDNF* gene were from Timmusk and colleagues, which identified in rats four 5’ exons linked to separate promoters and one 3’ exon encoding the preproBDNF protein [27,28,29]. These four *BDNF* promoters owned multiple points of *BDNF* mRNA regulation and suggested an activity-dependent regulation [28,29,30,31]. Further studies, published in 2007, clarified that *BDNF* has a complex gene structure with 11 different exons in humans, nine different exons in rodents, and nine alternative promoters for both groups [32,33]. Despite this complexity, the coding sequence is located in exon IX in both human and rodents. The latter includes the common sequence that encodes for the proBDNF protein. All other exons are untranslated regions with a start codon present in exons I, VII, VIII, and IX of the human *BDNF* gene. Exon IX is present in all BDNF mRNA isoforms. It is supposed that the nine alternative promoters can regulate the complex spatio-temporal expression of *BDNF* gene and allow BDNF to respond to a greater variety of stimuli. For instance, in human brain tissues, all exons are expressed, but to different degrees and in different brain structures [33]. Human heart tissue, instead, expresses high levels of *BDNF* isoforms containing exon IV and exon IX [33].

Currently, two *BDNF* promoters, promoter I and promoter IV, have been well characterized for their response after the activation of the L-type voltage gated calcium channel (L-VGCC) or the n-methyl-d-aspartate (NMDA) receptor. Activation of L-VGCC and NMDA receptors mediate intracellular Ca^2+^-signaling and regulate several aspects of brain functions (for review [34,35]). Promoter I is more responsive to neuronal activity and induces activity-dependent expression of BDNF in vitro and in vivo. It contains calcium-responsive elements (CaREs) and cyclic adenosine monophosphate (cAMP)/calcium response element (CRE) [29,36,37,38]. Deletion of CRE or overexpression of dominant negative of CREB (cAMP-response element-binding protein) significantly impairs rat BDNF promoter I response to neuronal depolarization [38]. Human BDNF promoter I is similar to rat promoter, since an orthologous CRE-like element is also present [39]. However, mutation of this site did not affect human *BDNF* promoter I response to depolarization [39]. Human BDNF promoter I also contains an activator protein 1 (AP1) -like element and an asymmetric E-box-like element [39]. Mutation in E-box-like element reduces human *BDNF* promoter I induction, impairing the response to neuronal depolarization [39].

Another highly characterized *BDNF* promoter is the *BDNF* promoter IV that contributes significantly to activity-dependent *BDNF* transcription. Human and rat *BDNF* promoter IV are similar. In this promoter, three CaREs and three other regulatory elements involved in regulating rat *BDNF* promoter response to NMDA receptor activation have been identified [40]. NMDA receptor activation is capable of triggering *BDNF* exon IV transcription through a protein-signaling cascade requiring extracellular signal-regulated kinase (ERK), Ca^2+^/calmodulin-dependent protein kinase (CaMK) II/IV, phosphoinositide 3-kinases (PI3K), and phospholipase C (PLC). *BDNF* exon IV expression also seems capable of further stimulating its own expression through TrkB activation [41]. Additionally to the CaREs, two positive regulators have been identified: the NF-kB (nuclear factor kappa-light-chain-enhancer of activated B cells) [42] and NFAT (nuclear factor of activated T-cells) binding sites [30,43]. In contrast to these positive regulators, *BDNF* promoter IV also contains a negative regulatory element, the class B E-box. This is a binding site for a basic helix-loop-helix protein, BHLHB2, a suppressor of the bHLH gene superfamily [43,44]. NMDA treatment is able to remove BHLHB2 binding to the E-box and to increase rat *BDNF* promoter IV activity [43,44]. Disruption of *BDNF* promoter IV in mice significantly reduced the number of parvalbumin GABAergic neurons in the prefrontal cortex and impaired GABAergic activity [45]. These mice displayed depression-like behavior such as anhedonia-like behavior and increased latency to escape in the learned helplessness test [45]. Further evidence suggests a relationship between stress exposure and epigenetic regulation of *BDNF* promoter IV with the development of psychiatric disorders. Specifically, changes in *BDNF* promoter IV methylation levels are implicated in depression [46,47]. Preliminary evidence has demonstrated that patients with major depressive disorder (MDD) present a hypomethylation of the CpG-87 site of the promoter IV region of *BDNF* gene and are less likely to benefit from antidepressants [47,48]. In addition, *BDNF* disruption from promoter IV-derived transcripts impairs fear expression in mice, suggesting that cells expressing *BDNF* from promoter IV critically regulate hippocampal-prefrontal plasticity during fear memory [49,50].

### 2.2. miRNAs and BDNF

MicroRNAs (miRNAs) are a class of evolutionary conserved small non-coding single-strand RNA molecules, 18–25 nucleotide long, able to bind to 3′ untranslated regions (3′ UTR) of target mRNAs and promote their degradation or suppress their translation into proteins. MiRNAs are expressed abundantly within the nervous system in a tissue-specific manner and are crucial players in several biological processes, including neurogenesis, neuronal maturation, synapse formation, axon guidance, neurite outgrowth and neuronal plasticity [51,52,53]. Accumulating data indicate that synthesis of BDNF may be affected by miRNAs, indeed, a regulatory negative feedback loop between BDNF and miRNAs exists. That is, while BDNF treatment stimulates neuronal miRNAs expression, miRNAs generally inhibit the expression of BDNF [54]. This negative feedback loop is maintained in a state of equilibrium in normal cells. Alterations in miRNAs or in BDNF contribute to the pathogenic mechanisms involved in neurodegenerative diseases or neuropsychiatric disorders.

A number of recent studies, obtained from high throughput sequencing screening of different brain regions or from neurological disorders, have identified seven miRNAs (miR-15a, miR-206, miR-155-5p, miR-16, miR-103-3p, miR-330-3p, Let-7a-3p) correlated with BDNF [55]. Previous data published by Schratt et al. reported the involvement of miR-134 in BDNF-regulated dendritic spine size in hippocampal neurons. They demonstrate that miR-134 negatively regulates the spine size via repressing the translation of LIM kinase 1, which is known to regulate dendritic structures. BDNF is able to relieve the inhibition of LIM kinase 1 translation, and in this manner contribute to synaptic development, maturation and/or plasticity [56]. Recently, Baby et al. [57] found that miR-134 mediates post-transcriptional regulation of CREB1 and BDNF, as previously described by Gao et al. [58], who demonstrated that mutant mice lacking Sirtuin 1 (SIRT1) catalytic activity shows reduction in both CREB and BDNF proteins and upregulation of miR-134. Thus, higher levels of miR-134 negatively regulate synaptic plasticity [58]. MiR-134-mediated post-transcriptional regulation of CREB1 and BDNF prevents cognitive deficits in chronic unpredicted mild stress model (CUMS) [59]. At the same time, data published by Xin and coworkers [60] demonstrate that miR-202-3p silencing reduces the damage to hippocampal nerve in CUMS rats through the upregulation of BNDF expression. miRNAs could be an effective target also for the treatment of depression. Recent data demonstrate that miR-124-mediated post-transcriptional regulation of CREB1 and BDNF can improve depression-like behavior in a rat model [61]. Instead, miR-153 through the inhibition of activation of the JAK-STAT signaling pathway improves BDNF expression and influences the proliferative ability of hippocampal neurons in autistic mice [62].

In vitro studies have also allowed analysis of the effect of neurotoxins or anesthetic agents on miRNAs expression and in turn, on BDNF expression. For instance, differentiated PC12 cells, treated with 1-methyl-4-phenylpyridinium (MPP), show upregulation of miR-34a, miR-141, and miR-9, suggesting that perturbed expression of them may contribute to Parkinson’s disease (PD)-related pathogenic processes, probably by affecting the expression of B-cell lymphoma 2 (BCL2), BDNF, and SIRT1 as potential targets [63].

Instead, studies in vitro on embryonic stem cell-derived neurons demonstrated that inhibition of miR-375 and miR-107 ameliorates ketamine-induced neurotoxicity via inverse regulation of the BDNF gene [64,65].

In summary, understanding the different functions of the various BDNF transcripts, the modulation of the expression of specific exons, and investigating the function of the BDNF-related miRNAs may represent a promising strategy to restore enduring changes in gene expression in response, for example, to environmental insults. This, in turn, might open new therapeutic perspectives for the treatment of neurodegenerative and neuropsychiatric disorders.

### 2.3. Biology of BDNF

Synthesis and maturation of BDNF is a multistage process, involving the formation of several precursor isoforms. The BDNF protein, discovered in 1982 [66], is a highly conserved protein of 247 amino acids, synthesized and folded in the endoplasmic reticulum as preproBDNF (32–35 kDa). Upon translocation to the Golgi apparatus, the signal sequence of the preregion is rapidly cleaved, and the isoform proBDNF (28–32 kDa) is generated [67]. The proBDNF is further cleaved to reach the mature isoform (mBDNF, 13 kDa) [67,68]. Intracellular proteolytic cleavage of proBDNF may occur by the subtilisin-kexin family of endoproteases such as furin, or in intracellular vesicles by convertases [69,70] (Figure 1).

Extracellular cleavage of proBDNF is determined by plasmin [71] and matrix metalloproteases 2 and 9 (MMP2 and MMP9) [72,73]. Depending on the cell type, BDNF can be secreted in a constitutive or activity-dependent manner [74]. In neuronal cells, both proBDNF and mBDNF are released following cell membrane depolarization [75,76,77]. The balance of proBDNF and mBDNF depends on the particular stages of brain development and regions. In the early postnatal period, the concentration of proBDNF is higher and may be considered as an important factor modulating brain function; while mBDNF prevails in adulthood and is important for processes occurring in adulthood, such as neuroprotection and synaptic plasticity [78]. Both proBDNF and mBDNF are active, eliciting opposing effects via the p75 neurotrophin receptor (p75NTR), a member of the tumor necrosis factor (TNF) receptor family and TrkB receptor, respectively. In resting form, both types of receptor are located in the membrane of intracellular vesicles. Stimulation with cAMP, Ca^2+^, or electrical impulse initiates their transfer and fusion with the cellular membrane [79,80].

The mature domain of proBDNF interacts preferentially with p75NTR, mediating synaptic pruning in the prenatal brain [81]. ProBDNF, through its pro-domain, can also interact with the sortilin receptor or other vacuolar protein sorting 10 protein (Vps10p) (Figure 2). Thus, proBDNF binding to specific receptors triggers signaling pathways, which can determine neuronal fate via promoting their death or survival [82,83]. The proBDNF/p75NTR/sortilin binding complex initiates signaling cascades leading to the activation of c-Jun amino terminal kinase (JNK). This pathway is involved in neuronal apoptosis [82,83]. High levels of p75NTR expression are detected during brain development and post-traumatic recovery [84]. When mature domain of BDNF binds to p75NTR, the RIP2 (serine/threonine-protein kinase 2)/TRAF6 (tumor necrosis factor receptor associated factor 6)-mediated pathway is initiated, which leads to NF-kB activation [82,85]. The activation of NF-kB promotes neuronal survival and maintenance during brain development [85]. In addition, p75NTR interacts also with the Ras homologous (Rho) protein family. This pathway is reported to regulate neuronal growth cone development and motility [85].

mBDNF binds with the high-affinity TrkB receptor, the receptor dimerizes, and the intracellular tyrosine residues are autophosphorylated [86]. Phosphorylated-TrkB activates several enzymes: PI3K, mitogen-activated protein kinase (MAPK), PLC-γ, and guanosine triphosphate hydrolases (GTP-ases) of the Rho gene family [87,88,89]. mBDNF-TrkB-signaling pathways regulate multiple events, such as apoptosis and survival of neurons [90,91,92], dendritic growth [93,94,95,96], spine maturation and stabilization, development of synapses [96,97,98], learning- and memory-processes-dependent synaptic plasticity [99,100].

PI3K/Akt-related pathway exerts antiapoptotic and pro-survival activity and modulates NMDA receptor-dependent synaptic plasticity [101,102,103]. The PI3K/Akt/mTOR cascade enhances dendritic growth and branching through regulation of protein synthesis and cytoskeleton development [104,105].

The MAPK/Ras-signaling cascade regulates protein synthesis during neuronal differentiation [85] and is also required for the activation of ERK 1/2 and CREB [106,107]. This pathway is crucial not only for early response gene expression (e.g., c-Fos), but also for cytoskeleton protein synthesis (e.g., Arc and cypin) [87], as well as dendritic growth and branching in hippocampal neurons [94,108].

The PLC-γ-dependent pathway evokes activation of CAM kinase and protein kinase C (PKC), which subsequently increases the 1,2-diacylglycerol (DAG) and Ca^2+^ ion concentrations [89]. The PKC-dependent pathway is reported to enhance synaptic plasticity [85] (Figure 2).

In summary, the specific role of BDNF in the regulation of numerous brain physiological processes depends on the interaction of its isoforms with different types of receptors. This, in turn, elicits the activation of signaling pathways that are critical for processes of brain development, synaptic plasticity, and protection and/or regeneration after damage. Perturbation of the BDNF synthesis, resulting in dysfunctions of its signaling cascades, may be responsible for triggering several pathological processes.

### 2.4. The Human BDNF Variant Val66Met

BDNF level in the peripheral tissues, brain, and blood may be also affected by gene polymorphism. The pro-domain of BDNF is the locus of a functional human BDNF polymorphism (SNPs) Val66Met, also known as rs6265 or G196Apolymorphism [109]. This point mutation causes a substitution of Valine (Val) to Methionine (Met) at codon 66 (Val66Met) in the pro-domain of *BDNF* (Figure 1). The Val66Met polymorphism does not exist in the mouse or other model organisms. Thus, multiple studies aim to mimic the function of BDNF Val66Met in cellular models or in genetically engineered mouse models. The BDNF Val66Met variant was first identified in the late 1990s and in 2002 the first two genetic studies investigating the BDNF Val66Met polymorphism in the pathogenesis of neurodegenerative disease were published [110,111]. The functionality of BDNF Val66Met variant was only confirmed in 2003 [112], where BDNF Val66Met polymorphism was shown to disrupt the episodic memory in humans. In addition, Egan et al. [112] also demonstrated that in hippocampal cultures BDNF Val66Met polymorphism did not alter BDNF expression per se, but the perisomatic localization of BDNF. Then, in 2005 it was discovered that the BDNF Val66Met substitution also disrupts the sortilin-binding site, impairing activity-mediated secretion of BDNF [113]. Likewise, the BDNF Val66Met substitution also disrupts the translin-binding site, which impairs dendritic targeting of BDNF mRNA [114]. Thus, the principle molecular mechanism associated with the BDNF Val66Met polymorphism is the deficient activity-dependent release of BDNF, which consequently impacts the efficiency of BDNF-TrkB signaling [113]. Following the demonstration that this SNP was functionally relevant over the past 18 years, more than 1700 studies have investigated the effects of this polymorphism on brain function in health, as well as in diseases, particularly in neuropsychiatric disorders [115,116]. The BDNF Val66Met polymorphism has been associated with cerebral cortex plasticity [117,118], with gray matter structures [119,120], or white matter integrities and structural networks [121,122]. More specifically, BDNF Val66Met polymorphism is associated with cognitive processes [112,123,124,125,126,127], and cognitive impairment in neurodegenerative disease, such as Parkinson’s disease (PD) [128,129] and AD [130,131], and even more with several brain disorders, including MDD and bipolar disorder [132,133,134,135,136,137], epilepsy [138,139,140], schizophrenia [125,141,142,143,144], aging and dementia [145] and stroke [117,146,147]. Met66, but not Val66, BDNF pro-domain can induce the growth cone retraction in young hippocampal neurons [148]. Although many studies have demonstrated the possible genetic effects of this BDNF polymorphism in diseases or brain function, other articles have failed to replicate the findings. The discrepancies of BDNF Val66Met genetic studies may result from many factors such as environmental factors, ethnicity, age, and sex.

## 3. Neuroplasticity in MDD: The Effects of Antidepressant Therapies

### 3.1. Major Depressive Disorder

Major depressive disorder is one of the most prevalent and debilitating psychiatric disorders with high impact on the quality of life and negative effects on mood, behavior, and cognition [149]. Over the past few decades, several mechanisms have been investigated in the pathophysiology of MDD, including altered serotonergic, noradrenergic, dopaminergic, and glutamatergic systems, increased inflammation, hypothalamic-pituitary-adrenal axis abnormalities, vascular changes, and decreased neurogenesis and neuroplasticity. In particular, a decrease in serotonergic neurotransmission is regarded as the main etiopathogenetic mechanism occurring in depressed patients. Thus, the most common drugs used to treat MDD are serotonin reuptake inhibitor (SSRI) that block SERT and thus increase serotonin in the raphe nucleus at post-synapse. Therefore, a misbalance in the serotonin production and/or release is believed to play a central role in determining MDD. This led to finding, by means of genetic, proteomic and pharmacological tools, molecules able to increase the expression of serotonin in neurons by modulating neural genes or proteins [150,151,152]. Among these molecules, TPH2, the rate-limiting enzyme responsible for brain serotonin biosynthesis, plays a crucial role and is amenable of genetic and pharmacological manipulation [153,154]. Nevertheless, in 1997 Duman and Nestler formulated the neurotrophin hypothesis of depression [155]. This theory is now supported by studies demonstrating a decrease in BDNF mRNA and protein levels in postmortem critical regions, such as the hippocampus, prefrontal cortex and amygdala, in patients with MDD compared to controls.

### 3.2. BDNF and Neuronal Plasticity

Brain development occurs through coordinated processes of neuro- and gliogenesis, formation of neuronal projections and synaptogenesis, and programmed cell death and elimination of improperly formed connections, together resulting in the formation of the functionally and morphologically adjusted structure of the adult brain [156,157]. Neuroplasticity or brain plasticity is the ability of the nervous system to reorganize its structure, function, and connections in response to extrinsic or intrinsic stimuli [158]. Neuronal plasticity in rodents has been well-documented during the last decades, whereas neuroplasticity in the human brain largely remains indirect, mostly because of methodological limitations as well as ethical constraints. Neuronal plasticity includes different mechanisms excellently reviewed by Castren [159]. One of these is the neurogenesis, i.e., the formation of newborn neurons in proliferative areas. There is solid evidence that neurogenesis occurs in the adult mammalian brain. In rodent adult brains, neurogenesis is mainly restricted to the subventricular zone and the subgranular zone of the dentate gyrus in the hippocampus and olfactory bulbs [156]. An accumulating body of evidence indicates that BDNF is involved in the regulation of migration of neuronal progenitors along the rostral migratory stream and neuronal settlement in the olfactory bulb [160] and also acts during the later stages of neurogenesis [161,162].

Neuronal plasticity is extensively studied during critical periods, a time window during the early phase of brain development, when neuronal circuits are noticeably sensitive to being shaped by external stimuli and experience, producing permanent and large-scale changes to neural circuits. The same circuits can be shaped by experience later in life, but to a lesser degree. After the ending of critical periods, neuronal plasticity and changes in network structure are more restricted. However, recent data indicate that several drugs used for the treatment of neuropsychiatric disorders can directly induce plasticity and reactivate a critical period-like plasticity in the adult brain. The first functional evidence for the role of neurotrophins in plasticity was obtained in the visual cortex. The observation that BDNF synthesis in the visual cortex is regulated by visual stimulation made BDNF the prime candidate for this activity-dependent regulated factor [163,164,165]. In transgenic mice with early overexpression of BDNF, an accelerated onset and end of the critical period and precocious maturation of inhibitory circuits was observed. Conversely, mice raised in the dark and resulting in lower levels of BDNF showed a delayed visual plasticity [166,167]. In addition, a disruption in the binding between promoter regions of BDNF exon IV and cAMP response element-binding protein (CREB) results in decreased inhibitory input [168], which impairs the critical period plasticity.

### 3.3. BDNF and Synaptic Plasticity

Another mechanism involved in neuronal plasticity is the modification of mature neuronal morphology, involving axonal and dendritic arborization and pruning, an increase in spine density, and synaptogenesis [169]. Epigenetic mechanisms involved in the transcriptional regulation of genes also can contribute to synaptic plasticity. Several in vitro and in vivo studies analyzed the effects of BDNF on plasticity. Cazorla et al. proved that 48 h of BDNF stimulation in PC12 cells, transfected with TrkB, increased neurite outgrowth compared to the non-treated cells [170]. Interestingly, BDNF stimulation was able to promote dendritic outgrowth and spine formation [171,172] in primary hippocampal cells grown in B27-deprived medium. This neuroplastic effect is probably achieved through the activation of intracellular-signaling cascades [173,174]. Recent data suggest that intracellular overexpression of BDNF in hippocampal developing neurons induces maturation of excitatory and inhibitory synapses, with respect to exogenous application of BDNF [175]. BDNF mice lacking BDNF die during the second postnatal week [176] and BDNF deficit causes inhibition of dendritic arborization [92,177] and reduction of expression of genes functionally related to vesicular trafficking and synaptic communication [178]. Instead, heterozygous BDNF mice survive into adulthood and BDNF is required for several forms of LTP, the main mechanism mediating plasticity [179]. At morphological level, these mice display a specific hippocampal volume reduction [180] similar to that observed in heterozygous TrkB mice [181,182], but in contrast to p75NTR-deficient mice [183]. These findings suggest a link between hippocampal volume and BDNF-mediated TrkB signaling [181,182]. Over the last years, BDNF has been extensively studied as an important regulator of synaptic transmission and LTP in the hippocampus and in other brain regions. The effects of BDNF in LTP are mediated by TrkB receptors. In particular, in the hippocampus the neurotrophin is thought to act at pre- and post-synaptic levels, modulating synaptic efficacy either by changes in pre-synaptic transmitter release, or by increased post-synaptic transmitter sensitivity (see e.g., [15,16]) to induce a long-lasting increase in synaptic plasticity. This depends on individual circumstances. Thus, BDNF can be: (i) either, a mediator or a modulator of synaptic plasticity, (ii) both, a neurotransmitter that acts both at pre- and post-synaptic level simultaneously at the same individual synapse. Recent data published from Lin et al. revealed that in CA3 or CA1 regions anterograde BDNF-TrkB signaling is involved in LTP induction, while anterograde and retrograde BDNF-TrkB signaling contributes to LTP maintenance. BDNF in both pre-synaptic and post-synaptic terminals modulate basal neurotransmission and pre-synaptic TrkB, probably regulating pre-synaptic release [184]. In addition, it has also been shown that BDNF regulates the transport of mRNAs along dendrites and their translation at the synapse. These processes occur by modulating the initiation and elongation phases of protein synthesis, and by acting on specific miRNAs [100]. Local protein synthesis responds with rapid and subtle modulation of the proteome to remodel the synaptic regions in response to stimuli [185]. Protein turnover is required for synaptic plasticity, and BDNF-signaling has been also described as a crucial regulator for maintaining the baseline autophagic activity in the brain. BDNF deficiency causes an uncontrolled rise in autophagic degradation [186].

BDNF is one of the most studied synaptic molecules that efficiently modify synaptic strength and can act as a mediator, modulator, or instructor of synaptic plasticity. Specific changes in dendritic spines, as well as in adult hippocampal neurogenesis, can be correlated to several forms of learning and memory. BDNF is one of the most inspiring molecules to better understand the disadvantageous synaptic learning underlying the etiology of depression, accompanied by declines in the rate of adult neurogenesis and in spine densities [181].

### 3.4. BDNF in Depressed Patients

BDNF protein and TrkB receptor are detectable in several non-neuronal tissues, including endothelial cells [17,18], cardiomyocytes [19], vascular smooth muscle cells [17], leukocytes [20], megakaryocytes [19], and platelets [21,22]. Serum BDNF has been clearly demonstrated to originate from the progenitors of platelets [21]. Platelets are the major source of peripheral BDNF and are important for storing the BDNF secreted from other tissues [187]. Over the last years, there has been a great interest in peripheral BDNF measures in relation to psychiatric illness. It has been studied as biomarker reflecting these disorders [188,189]. However, there is no evidence that serum BDNF is related to brain BDNF and neuroplasticity. Nevertheless, the low serum concentration of BDNF has often been associated with the pathophysiology of MDD [190,191,192]. An aspect to consider is if the serum BDNF levels are dependent on the release of BDNF from platelets [193]. The significance of the lower BDNF levels in depression is currently unclear. The temporal correlation between serum BDNF levels and the antidepressant effect seems to be indirect: ketamine and electroconvulsive shock treatment increase serum BDNF levels only gradually, while their antidepressant effect appears quickly [194]. There are two studies that directly observed a reduction of BDNF levels in platelets of patients with MDD [195,196]. Another study showed that BDNF levels of platelet were significantly decreased compared to the controls. In this study, the BDNF levels were normalized compared to control with SSRIs treatment [197]. Taken together, these studies strongly suggest that changes in serum BDNF levels reflect altered BDNF release from blood platelets. Thus, given the similarities in the regulation of BDNF synthesis between megakaryocytes and neurons, there may be parallels between the brain, BDNF in serum, and release. Nevertheless, within the CNS a reduction in BDNF and TrkB expression has been reported in the hippocampus and prefrontal cortex of post-mortem brain tissues of suicide victims [198,199]. In addition, several meta-analyses data confirm the association of the Val66Met polymorphism with an increase of susceptibility to develop mood disorders [200,201,202]. Finally, a recent paper showed that subjects with the Met allele of the BDNF gene are more likely to develop depression [134].

A disruption in serotonin signaling in the brain is also believed to be involved in the pathophysiology of depression. Changes in synaptic serotonin levels and receptor levels are coupled with altered synaptic plasticity and neurogenesis [203,204]. It has been proposed that chronic treatment with conventional antidepressants, such as SSRIs, but not acute administration increases neurogenesis [205,206,207] and selective SSRIs might reactivate serotonin’s ability to mediate developmental plasticity. BDNF acts as a modulator of the 5-HT system and vice versa, acting as the link between the antidepressant drug and the neuroplastic changes. Close molecular connections between serotonin receptors and neurotrophic proteins such as BDNF and intracellular signaling cascades are responsible for cytoskeletal rearrangement [169,208,209,210,211]. Thus, dysregulation in 5-HT–BDNF interaction may be responsible for the development of neuropsychiatric and behavioral abnormalities [212].

Understanding the function of the members of the BDNF system in response to the challenges of the environment and the interaction with different 5-HT receptors in health and disease will lead to new classes of drugs that could be used in therapy for psychiatric and neurodegenerative disorders.

### 3.5. Effect of Antidepressant Therapies on Plasticity BDNF-Mediated

#### 3.5.1. BDNF and Antidepressant Treatments

Multiple lines of evidence suggest that antidepressant treatments increase BDNF mRNA and protein levels in the cerebral cortex and hippocampus (for review see [213,214]). This increase is partly due to a reduction of histone acetylation in the *BDNF* promoter regions. The involvement of BDNF in the efficacy of antidepressant treatments has mainly been demonstrated in rodent models. It has been demonstrated that all pharmacological classes of clinical antidepressants increase TrkB autophosphorylation and signaling in the hippocampus and forebrain, effects observed within hours after the administration of the drug [203,215]. Similar results in BDNF mRNA and TrkB phosphorylation have been observed after acute treatment with ketamine [216,217,218,219]. In rodents, injection of BDNF in the hippocampus reduces depression-like behavior [220], in contrast injection of BDNF into the nucleus accumbens or ventral tegmental area promotes depressive effects [221], demonstrating the network-dependent effect of BDNF in mood regulation. Interestingly, conditional knockout of BDNF in forebrain regions increases depressive behavior in females, but not in male mice [222], and blocks the effects of antidepressants desipramine or ketamine [216,223]. Similarly, conditional deletion of TrkB in dentate gyrus or inhibition of TrkB signaling by a dominant-negative TrkB receptor blocks the effects of antidepressants [224,225]. In addition, mice with Val66Met polymorphism are insensitive to antidepressants [226]. Recent evidence demonstrates that the antidepressant effects of GLYX-13, a novel glutamatergic compound that acts as an NMDA modulator with glycine-like partial agonist properties, are blocked by intra-medial prefrontal cortex infusion of an anti-BDNF antibody or in mice with a knock-in of the BDNF Val66Met allele. Pharmacological inhibition of BDNF-TrkB signaling or L-type voltage-dependent Ca^2+^ channels (VDCCs) blocks the antidepressant behavioral actions of GLYX-13 [227].

Taken together, these data suggest that BDNF serves as a transducer, acting as the link between the antidepressant drug and the neuroplastic changes that result in the improvement of depressive symptoms. 

#### 3.5.2. Beneficial Effects of Exercise on Plasticity: The Role of BDNF

Several lines of evidence suggest that exercise has beneficial effects on plasticity and BDNF could be a link between plasticity and physical activity. Although it has been proven that exercise in MDD patients reduced depressive symptoms [228,229,230], neuroplasticity per se has not yet been monitored in these patients. However, voluntary physical exercise, like an enriched environment, increases expression of BDNF in the hippocampus [8], as well as hippocampal neurogenesis [9] and this could improve brain function by enhancing plasticity, cognition, learning, and memory [12,13,14]. Physical exercise is one particularly effective strategy for increasing circulating levels of BDNF [10,11]. It has repeatedly been demonstrated that an acute bout of aerobic exercise transiently increases both serum and plasma BDNF in an intensity-dependent manner [10,11]. Exercise increases the release of BDNF from the human brain [231,232] suggesting that exercise also mediates central BDNF production in humans. It has been suggested that miR-34a potentially can also mediate changes in BDNF expression and may reflect the decrease in performance after overtraining [233].

Multiple studies suggest that BDNF has a dominant role in mediating the effects of physical activity on cognitive changes [234]. It has been shown that three months of aerobic exercise training increases hippocampal volume in healthy individuals and in patients with schizophrenia by 12% and 16%, respectively [235]. The question whether exercise regulates muscle-derived circulating factors that can pass through the blood–brain barrier and stimulate BDNF production in the brain remains unclear. In 2016, Moon et al. show that the myokine cathepsin B (Ctsb) might be involved in mediating the exercise-induced improvement in hippocampal neurogenesis, memory, and learning [236]. Mice lacking Ctsb showed depression-like symptoms when they were forced to swim [236].

Other papers have demonstrated that exercise induces upregulation in skeletal muscle of PGC1α, a transcriptional co-activator of mitochondrial biogenesis and oxidative metabolism in brown adipose tissue and muscle. In muscle, the increase of PGC1α expression stimulates an upregulation of FNDC5, a membrane protein that is cleaved and secreted into the circulation as the myokine irisin [237]. FNDC5 cross the blood–brain barrier inducing BDNF expression in the hippocampus, in this way BDNF plays a role in neurogenesis and reward-related learning and motivation [238]. Current research has also shown that high intensity exercise increases peripheral lactate and BDNF levels; at the same time lactate infusion at rest can increase peripheral and central BDNF levels. Lactate and BDNF can induce neuroplasticity [239]. In addition, acute elevation of BDNF did not compensate for hypoxia-induced cognition impairment [240].

The identification of exercise-related factors that have a direct or indirect effect on brain function has the potential to highlight novel therapeutic targets for neurodegenerative diseases.

## 4. The Protective Role of BDNF on Neurodegeneration

Neurodegenerative diseases comprise a wide range of neurological diseases such as AD, PD, Huntington’s disease, and amyotrophic lateral sclerosis (ALS), characterized by the deterioration and then the death of selective nuclei of neurons in the brain or the spinal cord. They are chronic and progressive diseases, currently incurable and highly debilitating, causing a tremendous emotional and economic burden on patients, their families, and society. AD, the most frequent among neurodegenerative diseases, accounts for about 70% of dementia cases all over the world, that is about 35 million people. It is estimated to cost more than 480 billion euros each year throughout the world (Sources: OMS, EBC (European Brain Council)). Currently, no pharmacological treatment is available to cure or even significantly slow down the course of neurodegenerative diseases. For these reasons, experimental findings showing that physical exercise, exposure to an enriched environment, metabolic changes and nutritional and/or cognitive intervention, may exert a protective role on neurodegeneration either by delaying the onset and/or curbing the course of the disease, raise hope that these new tools might be useful also in clinical practice. BDNF appears to be crucial or, in some cases even essential, to mediate the neuroprotective effects of the above-mentioned environmental stimuli (Figure 3). In particular, as discussed above, it is well established that BDNF accounts for the hippocampal adult neurogenesis, which, in turn, can be stimulated by a number of conditions such as physical exercise, enriched environment, hormonal balance (i.e., steroid hormones such a cortisol and testosterone) and nutritional intervention (i.e., fasting, low-calorie intake, low-carb diet, selective nutrient intakes), capable of increasing the BDNF level [241].

### 4.1. The Protective Role of BDNF on Alzheimer’s Disease

A reduced level of BDNF has been found in patients affected by neurodegenerative diseases such as Parkinson’s, Huntington’s and Alzheimer’s disease as well as in mild cognitive impairment, the latter being a prodromal stage of AD, characterized by a slight decline of cognitive abilities including memory, thinking and judging skills [242,243,244,245]. In some instances, the levels of BDNF even correlate with the severity of the diseases, pointing towards a pathogenetic link between BDNF and AD [246]. Although there are some papers reporting an increase of BDNF in serum or in the post-mortem brain, this might be due either to compensatory mechanisms or its release from immune cells or pharmacological treatments known to raise BDNF (i.e., antidepressants) [241].

Recently, a very complex study explored the role of physical activity in a genetic mouse model of AD. This study provided the most compelling evidence of the relationship between physical activity, adult hippocampal neurogenesis, BDNF, and AD. This study elegantly confirms that adult hippocampal neurogenesis plays a pivotal role in brain resilience to AD. They manipulated with pharmacological and genetic tool neurogenesis as well as BDNF, clearly showing that physical exercise needs neurogenesis to protect the brain from AD and that BDNF is essential for such a protection. In addition, it provides evidence that adult hippocampal neurogenesis can counteract AD memory impairment, only in combination with BDNF, whereas if neurogenesis is experimentally blocked, BDNF does not exert beneficial effects. Finally, pharmacological increase of BDNF further ameliorates AD pathology [247]. Thus, agents that promote both BDNF signaling and neurogenesis might be the key to preventing or curing AD. As far as metabolism is concerned, it has been shown that intermitting fasting, by causing a transition from utilization of carbohydrate and glucose to a fatty acid and ketones source of energy (refer to as “G-K shift”) generates a number of beneficial cognitive, metabolic, and cardiovagal effects. BDNF is increased upon intermitting fasting and mediates at least part of these effects. Its increase is stimulated by the ketone body, β-hydroxybutyrate that inhibits histone deacetylases that repress BDNF promoters [248]. Recently, the role of BDNF and neuroprotection in the context of metabolism and fasting has been nicely reviewed by Mattson et al. [6].

### 4.2. BDNF and Ras-ERK-CREB Signaling in Alzheimer’s Disease

BDNF, as also discussed above, causes the activation of the Ras-ERK signaling cascade leading to the phosphorylation of CREB. Such a pathway exerts a well-known trophic and protective role on neuronal cells both in vitro and in vivo in a variety of neurodegenerative models, including AD, PD, and Huntington’s diseases. Nevertheless, it has become clear that the Ras-ERK pathway may also foster neurodegeneration or hamper the action of neurotrophic factors when activated by noxious stimuli as occurs for instance in PD and AD [249]. In particular, it has been shown in a number of different cellular models such as primary cortical rat neurons, rat B103 neuroblastoma cells, and A1 mouse mesencephalic cells, that APP and/or Aβ42 oligomer induces the activation of Ras-ERK and GSK-3 signaling, that, in turn, causes hyperphosphorylation of tau and APP at Thr668. The involvement of these molecular events in the pathogenesis AD is corroborated by the finding that activation of Ras-ERK and GSK-3 correlates with Aβ levels in the brain of AD patients [250,251]. Aberrant stimulation of Ras-ERK signaling forces neurons to enter the cell cycle as shown by the expression and nuclear accumulation of cyclin D1 and the subsequent G1/S progression. Since neurons lack functional cell cycle machinery, these events lead to cell death (i.e., mitotic catastrophe) instead of cell division. Interestingly, as clearly shown in the mouse model of familial AD APPswe/PS1ΔE9 mice, although ERK phosphorylation is enhanced compared to the wild type counterpart, it does not result in normal CREB phosphorylation. The impairment in CREB signaling parallels to impairment in a number of cognitive tests [252]. Therefore, in AD, BDNF downregulation is mediated by the impairment of CREB signaling caused by amyloid β [253] (Figure 3).

## 5. BDNF and Brain Cancer: An Unexpected Role. An Oncogene or a Tumor Suppressor?

The role of BDNF and its cognate receptor TrkB in cancer, including brain cancer, has been recognized for a long time [254]. In many types of cancers, BDNF and/or TrkB have been found expressed or in some cases over-expressed [255]. This is not surprising since growth factors, including neurotrophic factors, and their tyrosine kinase receptors have long been involved in tumors with different cell-dependent mechanisms, fostering proliferation, enhancing anti-apoptotic signaling, and making cells unresponsive to anti-proliferative stimuli [256]. The direct oncogenic activity of TrkB might also be due to the crosstalk with EGF receptors that together with its ligand is well-known to promote cell transformation. BDNF administration not only does phosphorylate TrkB but also EGFR [257]. In line with these observations, it has been recently shown that BDNF produced by glioblastoma (GBM) differentiated cells acts on GBM stem cells, fostering their growth through paracrine signaling [258].

However, recently another study showed that exposing mice to an enriched environment is able to decrease the growth of intracranial glioma, decreasing proliferation and invasion, and improving overall survival. Such an effect is achieved by means of both indirect and direct mechanisms. The former acts via natural killer cells of the innate immune system, whereas the latter utilizes BDNF stimulation of its truncated receptor TrkB.T1 on glioma cancer cells. BDNF binding the TrkB.T1 receptor signals to the Rho protein dissociation inhibitor (RhoGDI), the latter detaches from TrkB.T1 and binds to the small G protein RhoA, leading to its inhibition. The authors found that an enriched environment causes the synthesis of IL-15 and BDNF. When mice bearing the glioma and not housed in enriched environments were infused with BDNF, they reduced tumor size and macrophage infiltration. Thus, showing that at least in part, BDNF accounts for the oncolytic effect elicited by the enriched environment [259]. In a more recent study, the same group delved deeper into the mechanisms, finding that enriched environment changes glioma-associated myeloid cells. BDNF plays a central role by stimulating the production of IL-15 in microglia, which in turn stimulates the natural killer cells to produce IFN-γ. Natural killer cells were responsible for the switch to an oncolytic environment [260] (Figure 3).

Taken together, a scenario emerges where BDNF, acting on different cells is able to reorganize the brain microenvironment in such a way that it becomes resilient to neurodegeneration or oncolytic for tumors. In this regard, although supported by much more preliminary data, it seems that also other compounds might share these properties [261,262].

## 6. Conclusions

In this review, we discussed the role of BDNF in neurogenesis, differentiation, survival, synaptic plasticity, and transmission to reorganize the brain microenvironment. All BDNF gene products, such as proBDNF, mature BDNF, and even the isolated proBDNF domain, are known to exert functional activity. One of the most important features of BDNF is that it can act as a local, paracrine and/or autocrine factor, on both pre-synaptic and post-synaptic target sites. Here, we presented the contribution of altered BDNF signaling in the pathophysiology of brain diseases, including mental disorders (i.e., depression), neurodegenerative diseases, (i.e., Alzheimer’s disease), and brain tumor (i.e., glioblastoma). BDNF is one of the best-studied synaptic molecules that efficiently modify synaptic strength and it can act as a mediator, modulator, or instructor of synaptic plasticity. In neurons, the cellular processes that regulate the amount of both BDNF mRNA and protein, the changes in the efficiency of secretion, and transport of BDNF protein may affect synaptic function and cell survival. BDNF is one of the most inspiring molecules to better understand the disadvantageous synaptic learning underlying the etiology of depression, accompanied by the decline in the rate of adult neurogenesis and in spine densities. Furthermore, BDNF appears to exert a potent role in neuroprotection and/or brain regeneration by modulating signaling pathways such Ras-ERK-CREB, thus rendering neuronal cells resilient to neurodegeneration. Finally, BDNF appears to be crucial in the pathogenesis and development of brain tumors such as glioblastoma by reorganizing its microenvironment. Thus, understanding the physiologic and pathologic BDNF signaling and finding tools to modulate its expression (mRNA and/or protein) is a prerequisite for a potential BDNF-based therapy.

## Figures and Tables

**Figure 1 ijms-21-07777-f001:**
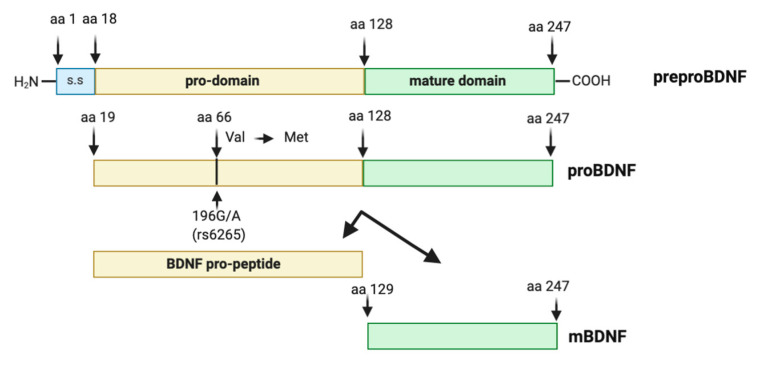
Brain-derived neurotrophic factor (BDNF) protein structure. The preproBDNF consists of three sequences: signal sequence (s.s), pro-domain, and mature domain. The intra- or extracellular cleavage of preproBDNF generates functionally active isoforms: BDNF pro-peptide and mature BDNF (mBDNF), each of which exhibits a characteristic affinity to a specific type of receptor. Arrowheads indicate known protease cleavage sites involved in the processing of mature BDNF. The position of the single nucleotide polymorphism (rs6265, Val66Met) and the substitution of valine (Val) in methionine (Met) at codon (aa) 66 in the human BDNF gene is indicated by an arrow.

**Figure 2 ijms-21-07777-f002:**
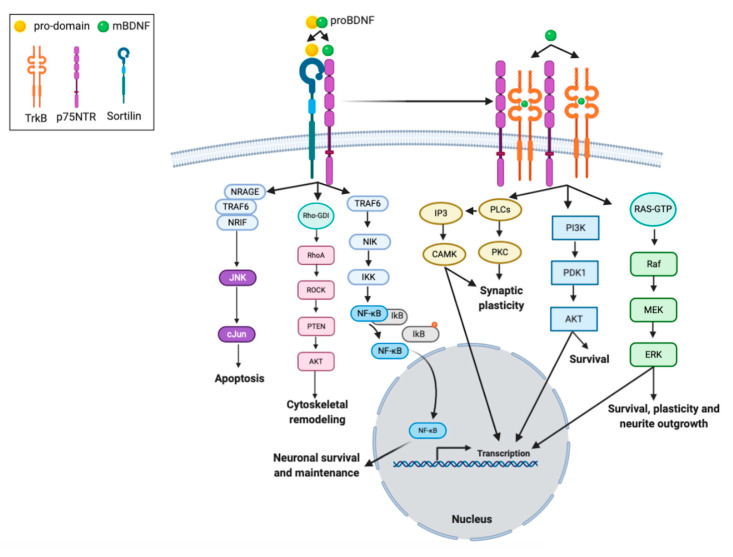
Intracellular signaling cascades activated by interaction of BDNF isoforms with its receptors. proBDNF and mBDNF bind to different receptors, respectively. The mBDNF isoform exhibits highest affinity for the tyrosine kinase B receptor (TrkB) receptor, which when stimulated undergoes homodimerization and autophosphorylation, but also binds the low affinity neurotrophin receptor p75NTR. The interaction between the TrkB receptor and the p75NTR receptor in a complex increases the ligand binding affinity to BDNF. Sortilin is considered a co-receptor for p75NTR. The proBDNF isoform, consisting of two sequences (pro-domain and mature domain), interacts with specific receptors, sortilin and p75NTR, respectively. The binding of proBDNF to a p75NTR/sortilin-complex induces signaling pathways that are specific for proBDNF. The binding of proBDNF in combination with sortilin causes the involvement of neurotrophin receptor-interacting factor (NRIF), tumor necrosis factor receptor-associated factor 6 (TRAF6), and neurotrophin receptor-interacting MAGE homologue (NRAGE) proteins. This pathway activates the JNK-associated pathway that promotes programmed cell death, or the receptor-interacting serine/threonine-protein kinase 2 (RIP2) /TRAF6-mediated pathway is initiated. Multi-subunit IκB kinase (IKK) phosphorylates (orange dot) the inhibitor of kB (IkB) protein, which results in dissociation of IkB from NF-κB. The activated nuclear factor kappa B (NF-kB) is then translocated into the nucleus where it binds to specific sequences of DNA and promotes neuronal survival and maintenance. In addition, p75NTR interacts with the Rho family of proteins, whose activation mediates the activity of Rho-associated protein kinase (ROCK), which subsequently leads to activation of the AKT pathway, involved in cytoskeletal remodeling. The mBDNF/TrkB receptor complex triggers signaling pathways associated with activation of phosphatidylinositol 3-kinase (PI3K), phospholipase C gamma (PLC-γ), and GTP-ases of the Rho family, involved in survival, plasticity and neurite outgrowth, transcription regulation, and synaptic plasticity.

**Figure 3 ijms-21-07777-f003:**
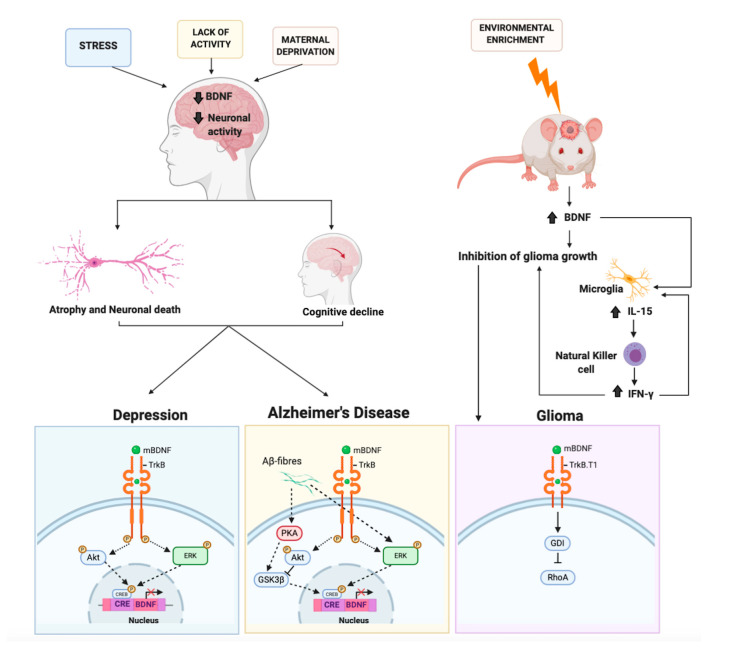
The molecular mechanisms mediated by BDNF involved in depression, Alzheimer’s disease, and glioma. External stimuli (stress, maternal deprivation, or lack of activity) causing epigenetic regulation processes can induce a reduction in BDNF expression level and in neuronal activity. This results in atrophy, neuronal death, and cognitive decline, which may contribute to depression or Alzheimer’s disease. In these pathologies the BDNF/TrkB signaling, which activates the downstream Akt and ERK signaling, is altered. Thus, such alterations cause an impairment of CREB signaling resulting in BDNF downregulation. The alteration of phosphorylation (P) inhibits (red X) the transcriptional machinery. In the mouse brain, environmental enrichment induces an increase of BDNF. BDNF, binding the truncated form of TrkB receptor (TrkB.T1), signals directly to the Rho protein dissociation inhibitor (GDI). The latter detaches from TrkB.T1 and binds to the small G protein RhoA, leading to an inhibition of glioma cell migration. BDNF stimulates also indirectly the production of IL-15 in microglia, which in turn stimulates the natural killer cells to produce IFN-γ. IFN-γ contributes to reducing glioma growth.

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
