# Peer review of "Neurotrophic Factor BDNF, Physiological Functions and Therapeutic Potential in Depression, Neurodegeneration and Brain Cancer"

_ijms, 2020, doi:10.3390/ijms21207777_

Round 1

Reviewer 1 Report

The review article contains new interesting and relevant data on the stated topic. However, an improvement in the writing style of the article is required and the article must be shown to a native speaker.
Notes to the review article.
1. In the introductory part, the amount of modern data should be increased (last 10 years). In the presented version, the number of modern works is less than 30%, it is necessary to increase to 50%.
2. After several sections: 2.1 BDNF transcripts, 2.2 miRNAs and BDNF, a final summary of these sections should be done. In these sections, you need to improve your writing style.
3. Sections 3.2 BDNF and neuronal plasticity and 3.3 BDNF and synaptic plasticity are well written and very informative. They should be used as a standard to improve the content of other sections of the review.
4. Show the manuscript to a native speaker.
5. In the summary, the wording of the basic content of the review article should be revised. Authors should make this section more informative, meaningful and understandable. It is also necessary to improve the writing style of this section and the article in general.

Author Response

Please find below, in red, detailed point-by-point responses to the reviewers’ comments.

Point 1: In the introductory part, the amount of modern data should be increased (last 10 years). In the presented version, the number of modern works is less than 30%, it is necessary to increase to 50%.

Response 1: We changed the introduction and we included modern references (lines 40-67). The percentage of modern work is now about 70%  (lines 32-86).

Point 2: After several sections: 2.1 BDNF transcripts, 2.2 miRNAs and BDNF, a final summary of these sections should be done. In these sections, you need to improve your writing style.

Response 2: We included a final summary after section 2.2 (lines 214-218) and the text is changed improving the writing style (lines 87-218).

Point 3: Sections 3.2 BDNF and neuronal plasticity and 3.3 BDNF and synaptic plasticity are well written and very informative. They should be used as a standard to improve the content of other sections of the review.

Response 3: We improved the writing style in the other sections of the review.

Point 4: Show the manuscript to a native speaker.

Response 4: A native speaker revised the manuscript.

Point 5: In the summary, the wording of the basic content of the review article should be revised. Authors should make this section more informative, meaningful and understandable. It is also necessary to improve the writing style of this section and the article in general.

Response 5: We have completely modified the summary (lines 738-767).

We would like to thank the reviewer for his constructive criticisms and insightful comments that helped us to improve our review.

Reviewer 2 Report

The authors have extensively described the role of BDNF in neuronal plasticity, depression, alzheimer's along with leaving the readers with its possible role in cancer. The references are sufficient and succinct. Please check the entire document for minor spell check and grammatical errors. 

Author Response

Response to Reviewer 2 Comments

Please find below, in red, detailed point-by-point responses to the reviewers’ comments.

Point 1: Please check the entire document for minor spell check and grammatical errors. 

Response 1: We revised the manuscript for minor spell check and grammatical errors.

We would like to thank the reviewer for his constructive criticisms and insightful comments that helped us to improve our review.

Reviewer 3 Report

In this manuscript the authors describe the  physiological function and therapeutic potential of BNDF in depression, neurodegeneration and brain cancer. The authors claim that BNDF signaling contribute to the development of several neurological disorders. While the manuscript is well-written and clearly understandable, I would suggest to include an additional figure highlighting those pathways which by the BNDF modulate the development of the diseases.

The authors describe BDNF Val66Met polymorphism is associated
with cognitive processes. What is the molecular signaling behind this association?

Author Response

Please find below, in red, detailed point-by-point responses to the reviewers’ comments.

Point 1: I would suggest to include an additional figure highlighting those pathways which by the BNDF modulate the development of the diseases.

Response 1: We included a new figure 3 and relative legend.   

Point 2: The authors describe BDNF Val66Met polymorphism is associated
with cognitive processes. What is the molecular signaling behind this association?

Response 2: We discussed the molecular signaling associate to BDNF Val66Met in lines 370-380.

We would like to thank the reviewer for his constructive criticisms and insightful comments that helped us to improve our review.

Round 2

Reviewer 1 Report

Having reviewed the revised version of the manuscript of the review article "Neurotrophic factor BDNF, physiological functions and therapeutic potential in depression, neurodegeneration and brain cancer", written by Luca Colucci-D’Amato *, Luisa Speranza, Floriana Volpicelli, i can conclude that after the revision, the manuscript has undergone a significant improvement in content and writing style. In this form, the manuscript can be recommended for publication in IJMS.